# Is Senolytic Therapy in Cardiovascular Diseases Ready for Translation to Clinics?

**DOI:** 10.3390/biom15040545

**Published:** 2025-04-08

**Authors:** Zhihong Yang, Duilio M. Potenza, Xiu-Fen Ming

**Affiliations:** Laboratory of Cardiovascular and Aging Research, Department of Endocrinology, Metabolism, and Cardiovascular System, Faculty of Science and Medicine, University of Fribourg, 1700 Fribourg, Switzerland; duilio.potenza@unifr.ch (D.M.P.); xiu-fen.ming@unifr.ch (X.-F.M.)

**Keywords:** aging, heart, atherosclerosis, senescent cells, senolytics

## Abstract

Aging is a predominant risk factor for cardiovascular diseases. There is evidence demonstrating that senescent cells not only play a significant role in organism aging but also contribute to the pathogenesis of cardiovascular diseases in younger ages. Encouraged by recent findings that the elimination of senescent cells by pharmacogenetic tools could slow down and even reverse organism aging in animal models, senolytic drugs have been developed, and the translation of results from basic research to clinical settings has been initiated. Because numerous studies in the literature show beneficial therapeutic effects of targeting senescent cells in cardiomyopathies associated with aging and ischemia/reperfusion and in atherosclerotic vascular disease, senolytic drugs are considered the next generation of therapies for cardiovascular disorders. However, recent studies have reported controversial results or detrimental effects caused by senolytic therapeutic approaches, including worsening of cardiac dysfunction, instability of atherosclerotic plaques, and even an increase in mortality in animal models, which challenges the translation of senolytic therapy into the clinical practice. This brief review article will focus on (1) analyzing and discussing the beneficial and detrimental effects of senolytic therapeutic approaches in cardiovascular diseases and cardiovascular aging and (2) future research directions and questions that are essential to understand the controversies and to translate preclinical results of senolytic therapies into clinical practice.

## 1. Introduction

Organism aging is an established predominant risk factor for the cardiovascular system, including a group of maladaptive structural and functional alterations of the heart and blood vessels, resulting in atherosclerotic vascular disease, cardiac arrythmia, ischemia/reperfusion injury, and heart failure [1]. Cardiovascular aging and disease development share common mechanisms, such as low-grade chronic inflammation (inflammaging) [2], genomic damage, telomere attrition, epigenetic alterations, altered proteostasis, deregulated nutrition sensing, mitochondrial dysfunction, stem cell exhaustion, cellular senescence, etc. [3,4]. These mechanisms or hallmarks can occur in all cell types, including cardiomyocytes, fibroblasts, immune cells, endothelial cells, and smooth muscle cells, in the heart and blood vessels [5,6]. Modulation of these mechanisms has been reported to be capable of not only affecting the lifespan but also cardiovascular disease development, as demonstrated at least in animal models [7]. Among other therapeutic strategies, targeting senescent cells through senolytic approaches either pharmacologically or genetically has opened new paths to rejuvenate or slow down the aging process of organisms and to treat age-related diseases, including cardiomyopathies and atherosclerotic vascular disease [8,9]. Although senolytic approaches provide promising results and are regarded as the next generation of therapies for cardiovascular diseases, several detrimental effects in the cardiovascular system are emerging and have been recognized recently, which represents a challenge for further understanding physio-pathological functions of cellular senescence and the development of senolytics in the context of the treatment of cardiovascular diseases. In this brief review article, we do not intend to discuss the pathophysiological role of senescent cells in aging and age-related chronic diseases but rather the current evidence regarding beneficial and detrimental effects and controversial findings of anti-senescence therapy in cardiovascular diseases, such as cardiomyopathies and atherosclerotic vascular disease. Within this context, future research questions or directions that are important for translating preclinical results of senolytics into clinical practice are discussed in more detail. 

## 2. Accumulation of Senescent Cells in Cardiac Dysfunction and Atherosclerosis 

Cellular senescence, first described by Hayflick and Moorhead in the 1960s in cultured fibroblasts [10], has been recognized as a cell stress response, which is characterized by irreversible cell cycle arrest and associated with multiple changes in gene expression and active metabolic functions, including, typically, the enhanced expression and secretion of multiple factors, known as senescence-associated secretory phenotype (SASP), which includes cytokines, chemokines, growth factors, metalloproteases, etc. [11]. Senescent cells exhibit an increased capability of producing reactive oxygen species (ROS) from mitochondria and other enzymatic sources, such as eNOS uncoupling in endothelial cells and other cell types of blood vessels [12,13,14,15]. The SASP phenotype is considered to play a crucial role in inflammaging and tissue/organ damage in an autocrine manner and through paracrine mechanisms, leading to altered cell–cell interactions in cardiovascular aging and disease development [9,11,16]. Cell senescence can be triggered by numerous stressors, such as excessive cell replication (replicative senescence), oncogene activation, DNA damage, oxidative stress, hypoxia, SASP factors, etc. (premature senescence) [17,18]. It is important to recognize that senescent cells exhibit elevated anti-apoptotic mechanisms, such as anti-apoptotic proteins of the Bcl2 family, which render the cells resistant to programmed cell death [19,20]. Also, because of this feature of senescent cells, senolytic drugs that inhibit the anti-apoptotic mechanisms are developed and have shown significant efficacy in eliminating senescent cells in vivo [21,22,23]. The effects of senolytics as therapeutics in aging and cardiovascular diseases are the focus of this article and discussed later. Regarding detailed mechanisms and markers of cellular senescence, readers are referred to several comprehensive recently published review articles [5,8,11].

There is convincing evidence demonstrating that senescent cells are accumulated in failing heart and age-associated cardiomyopathies [24,25,26] and also found in atherosclerotic plaques in the blood vessels [27,28]. It has been shown that all cell types in the heart, including cardiomyocytes, endothelial cells, fibroblasts, progenitor cells, and immune cells, can undergo senescence with aging and/or cardiovascular diseases [29]. Adult cardiomyocytes in humans are terminally differentiated, rarely dividing into post-mitotic cells, which are, however, capable of acquiring a senescent-like phenotype through stressors under physio-pathological conditions. For example, many stressors, including inflammation, oxidative stress, hypoxia/reoxygenation, ischemia/reperfusion, chemicals, and aging, can induce cardiomyocyte senescence, which occurs independently of telomere shortening, i.e., length-independent telomere damage [24,30,31,32,33,34,35]. Senescent cardiomyocytes show upregulation of p21^cip1^, p16^ink4^, and SASP and exhibit pro-fibrotic and pro-hypertrophic features in aging mouse models [24,25]. Besides cardiomyocytes, other cell types, such as endothelial cells, fibroblasts, and immune cells, could affect the biological function of cardiomyocytes via paracrine release of SASP factors if they become senescent or are activated [6,36]. It has been shown that vascular endothelial cells in aged animals and humans exhibit the senescence phenotype, including enhanced expression of cell cycle inhibitors, inflammatory adhesion molecules VCAM-1 and ICAM-1, eNOS uncoupling in vascular endothelial cells, and SASP [5,15,36,37]. An increase in number of senescent endothelial cells is also found in diseased conditions, such as atherosclerosis [27], diabetic vascular diseases [38,39], hypertension [40], and heart failure [37]. Moreover, fibroblast senescence is associated with healthy aging in the heart [5] and in myocardial infarction [41]. Finally, stem and/or progenitor cells in the heart or from bone marrows are reported to be exhausted in aging and myocardial infarction, which could also be partly due to stem/progenitor cell senescence [42,43,44]. 

In atherosclerotic plaque, a large number of senescent cells was found in animal and human samples [45,46,47,48]. This observation suggests that senescent cells play a role in atherogenesis or plaque development. Multiple senescent cell types, including endothelial cells, vascular smooth muscle cells, and immune cells, including foam cells, macrophages, and T-cells, are strongly associated and correlated with atherosclerosis [27,28,49,50,51,52]. It is logically thought that these senescent cells contribute to atherogenesis via autocrine and/or paracrine effects on the vascular inflammation process and that targeting senescent cells can reduce inflammation, inhibit atherosclerotic plaque development, and stabilize the plaque [16]. 

## 3. Anti-Senescence Therapy in Cardiovascular Diseases: Beneficial or Harmful?

### 3.1. Cardiac Aging and Cardiomyopathies

Anti-senescence therapy initiated as a therapeutic strategy for cardiovascular disease and age-associated cardiomyopathies is based on various studies showing the accumulation of senescent cardiomyocytes and other non-cardiomyocytes under pathological conditions, such as myocardial infarction, heart failure, and atherosclerosis, as discussed above. The anti-senescence therapeutic concept was further promoted by several seminal studies demonstrating reversal of the aging phenotype after removal of p16^ink4+^ senescent cells through genetic and pharmacological means. Indeed, in a transgenic mouse model in which the *cdkn2a* gene encoding p16^ink4^ was specifically knocked out in the cardiomyocyte population, improved cardiac function and significantly reduced scar size were observed after myocardial infarction, which is associated with reduced senescence-associated inflammation and decreased senescence-associated markers of the heart [53]. Elimination of senescent cells pharmacologically and genetically decreases the infarct size, improves cardiac function, and reduces cardiac fibrosis after ischemia/reperfusion and in aged animals [24,32]. This beneficial effect of the anti-senescence strategy seems to be supported in other cardiomyopathy models, such as doxorubicin-induced cardiomyopathy in rats in which p16^ink4+^ cardiomyocytes are increased [34]. Moreover, deletion of the p16^ink4+^ gene in cardiomyocytes significantly improved cardiac function and reduced cardiac scar size and SASP after myocardial infarction in a mouse model [53], demonstrating a role of senescent cardiomyocytes in contributing to pathological cardiac remodeling and heart dysfunction following myocardial infarction. Furthermore, it has been shown that the elimination of senescent cells in naturally aging mice or Hutchinson–Gilford progeria syndrome (HGPS) mouse models, a premature aging disorder caused by a mutation of the nuclear-membrane-associated protein lamin A called progerin, ameliorated DNA damage, improved cardiac and kidney function, and also prolonged the lifespan [54,55,56]. Treatment of chronologically aged mice, radiation-exposed mice, and progeroid mice with the senolytic drugs dasatinib and quercetin reduced senescent cell burden, improved cardiac function, carotid vascular function, and exercise capacity, and extended the lifespan of progeroid mice [57]. These results suggest that targeting senescent cells could exhibit beneficial effects on cardiovascular function either in aging or in pathological conditions. Although the underlying mechanisms are not clear, it is, however, presumable that senescent cells, including senescent cardiomyocytes, and other cell types in the heart compromise cardiac function via autocrine and paracrine effects. The beneficial effects of the anti-senescence strategy for cardiac remodeling and dysfunction have been reported in several studies with senolytic drugs in animal models. Navitoclax (ABT-263), a potent inhibitor of cell survival Bcl-2 family proteins, which is upregulated in senescence cells [21], seems able to selectively eliminate senescent cells, including senescent cardiomyocytes and other non-myocytes in the heart [58]. It has also been shown to attenuate angiotensin II-induced cardiac dysfunction, cardiac hypertrophy and fibrosis, and inflammation in a mouse model [59], an ischemia reperfusion injury model, and a myocardial infarction model [58,60]. 

Despite these beneficial effects, anti-senescence drug therapy has also had detrimental effects in heart disease. In the mouse myocardial infarction model and ischemic human myocardium, increased senescent cardiomyocytes and SASP are found [61]. Surprisingly, a tail vein injection of AAV9-Gata4-shRNA injection into mice attenuates SASP but aggravates heart dysfunction after myocardial infarction. A study suggests that some SASP factors may be able to improve postinfarction heart function [61,62]. The underlying mechanisms could be due to the observation that some of these SASP factors are beneficial in supporting tissue repair or regeneration [63,64]. Moreover, not all senescent cells are functionally equal. In a study using a liver injury and repair mouse model, Zhao et al. demonstrated that macrophages and endothelial cells (ECs) represent distinct senescent cell populations with different fates and functions during liver fibrosis and repair [65]. Clearance of p16^Ink4a+^ macrophages mitigates hepatocellular damage, whereas eliminating p16^Ink4a+^ endothelial cells aggravates liver injury [65]. Taking into consideration that macrophages and endothelial cells are important cell types that contribute to cardiovascular disease, one would ask whether this finding may explain some controversial results in heart disease, as mentioned above, and vascular atherosclerotic disease, as discussed later. In addition, the SASP factors could cause cardiac fibroblasts to become senescent and suppress fibroblast proliferation/expansion, resulting in less cardiac fibrosis [62,66]. On the other hand, induction of cardiac fibroblast senescence seems to protect the heart against age-related cardiac fibrosis and dysfunction [67], while inactivation of the premature senescence program in the aortic coarctation mouse model aggravates cardiac fibrosis and cardiac dysfunction [66]. 

### 3.2. Atherosclerotic Vascular Disease

Indeed, the genetic model designed to deplete p16^ink4+^ cells shows not only a reversal of the aging phenotype, including the heart aging phenotype, but also a reduction of atherosclerotic burden in mouse models [46,55]. Recently, vaccination therapy designed to eliminate Gpnmb-positive cells reduced the number of p19^ARF+^ cells, reversing atherosclerosis in a progeroid mouse model [68]. These results show the important contribution of senescent cells to the pathogenesis of atherogenesis. 

However, another study using bone marrow transplantation to selectively ablate bone-marrow-derived or vessel-wall-derived p16^ink4+^ cells or both in atherosclerotic mice did not show an effect on atherosclerosis and instead increased the expression of the inflammatory cytokines TNFα and IL18 [69]. Moreover, in a hypercholesterolemic mouse model, a combination of the two senolytics dasatinib and quercetin reduced aortic calcification and improved vasomotor responses but did not affect atherosclerotic burden [70]. Most surprisingly, a recent study by Karnewar and colleagues demonstrated that in the SMC (Myh11-CreERT2-eYFP) and EC (Cdh5-CreERT2-eYFP) lineage tracing, ApoE^−/−^ mice fed a Western diet for 18 weeks, followed by ABT-263 at 100 mg/kg/bw for 6 weeks or 50 mg/kg/bw for 9 weeks, no effects on lesion size were observed, but enhanced plaque instability and enhanced mortality by 50% were seen in the treatment groups [71], implicating the detrimental effects of the senolytic ABT-263 in advanced atherosclerosis at least in the mouse model. These detrimental effects are associated with a 90% reduction of plaque smooth muscle cells and a 60% increase in endothelial–mesenchymal transition (EndoMT). A 60% reduction in α-SMA^+^ fibrous cap thickness is also observed in ABT-263 treatment [71]. This is a surprising but important finding, suggesting that one must be more cautious with senolytic therapy in clinical settings. It is important to confirm this finding further with this or other senolytic drugs in other relevant animal models, as multiple clinical trials with senolytic drugs are currently ongoing in treating various diseases, including Alzheimer’s disease and chronic kidney disease [72]. The controversial results with senolytic therapy in cardiovascular diseases are not clear. We will thoroughly discuss these diverse findings and point out further research directions in the next section.

## 4. Future Research Questions

According to the current research in the literature, we could conclude that senescent cells exhibit two faces in cardiovascular disease and aging, i.e., they have either detrimental effects or beneficial effects. Although the reasons for these contradictory results are not clear, the following considerations may provide hints for explanation and point towards further research directions. First, it has been demonstrated that not all senescent cells have the same functions. For example, with the development of mouse models for genetic tracing and the manipulation of p16^ink4+^ cells, specific depletion of senescent cell types, such as endothelial cells and macrophages, and other cell types becomes possible. Using these mouse models, Zhao and colleagues demonstrate a distinct role of senescent macrophages and endothelial cells in CCL4-induced liver injury and fibrosis [65]. They show that the ablation of specific p16^ink4+^ senescent macrophages reduces liver fibrosis, whereas the ablation of senescent endothelial cells exacerbates fibrosis [65]. Whether these distinct functions of senescent macrophages and endothelial cells or other senescent cell types could be demonstrated in the pathogenesis of cardiovascular diseases requires further investigation. Moreover, senescent cells derived from the same cell type are not homogenous even in the same disease microenvironment. At least two groups of senescent cells might exist, i.e., pro-inflammatory tissue destructive and anti-inflammatory tissue reparative senescent cells, depending on the profile of SASP. Indeed, it has been demonstrated that senescent endothelial cells can acquire an anti-inflammatory phenotype in response to oxidative stress, hypoxia, and disturbed flow [73,74,75], which is mediated by caveolae-associated proteins [76]. Moreover, it seems that the functions of early senescent cells are different from those of deep persistent senescent cells [77], which could contribute to the different outcomes of senolytic therapy. 

Second, it therefore seems that the effects of senolytic therapy, whether beneficial or detrimental, are context-dependent in specific diseases. In cardiac diseases, e.g., myocardial infarction, heart failure, or age-related cardiac dysfunction, removal of senescent cardiomyocytes could be detrimental if survived or available cardiomyocytes under the condition are not adequate or not sufficient to support the pumping function of the heart. On the other hand, the elimination of senescent cardiomyocytes will be beneficial if the remaining cardiomyocytes are adequate to compensate for the heart pumping function and the detrimental paracrine effects of cardiomyocytes on other non-myocyte cells, such as fibroblasts, endothelial cells, and immune cells, could be removed. Importantly, this beneficial effect could be strengthened through the elimination of senescent endothelial cells and immune cells. If cardiac fibrosis is the major problem in cardiac dysfunction, the elimination of senescent fibroblasts could enhance fibroblast proliferation, resulting in more pronounced cardiac fibrosis, as we have discussed in the previous section and as has been shown in live fibrosis [78,79]. Therefore, the functions of senescence of various cell types in the heart under specific heart disease conditions must be better defined. In this context, targeting specific senescent cell types in heart disease would be essential. Furthermore, modulation instead of elimination of senescent cells, particularly cardiomyocytes, might be a better strategy to improve cardiac function in heart disease. It has been shown that in a transgenic mouse model that allows for specific inactivation of the p16^ink4^ gene (*cdkn2a*) in cardiomyocytes, cardiac dysfunction caused by myocardial infarction is significantly improved, and scar size is reduced [53]. In this context, SASP factors are diverse, and some of them are beneficial and represent a compensatory mechanism for repairing. Removing the detrimental ones while keeping the beneficial ones would be an option to regulate the local microenvironment for cardiomyopathies. The senescent cells, including senescent cardiomyocytes, endothelial cells, etc., would not be eliminated, but the functions could be improved or reversed by targeting SASP factors (senomorphics or senostatics). Indeed, our recent study shows that modulation of inflammaging through genetic inhibition of mitochondrial arginase-II, an enzyme that is upregulated in many cell types, including macrophages, endothelial cells, and fibroblasts, and promotes mitochondrial ROS production and chronic inflammation in aging and senescent cells, reduces cardiac aging phenotypes and makes the aging heart more resistant to ischemia and reperfusion injury [36]. As discussed, senescent cells secrete a plethora of SASP factors, including the pro-inflammatory IL-1α, IL-1β, IL-6, IL-8, IL-18, (TNF)-α, HMGB-1, MCP1, MIP-1a, and MIP-3a, coagulation-modulating factors, such as PAI-1 and TF-I, extracellular matrix remodelling enzymes, such as MMPs, and growth factors, such as VEGF, TGFb, etc. Different cell types in the cardiovascular system, including cardiomyocytes, endothelial cells, fibroblasts, and immune cells, reveal distinctive profiles of SASP factors that contribute to a complex of cell–cell interactions in the development of cardiovascular aging and diseases (for a detailed description and summary of SASP factors from different cell types in the heart and blood vessels, please refer to several comprehensive review articles) [6,80,81,82]. Inhibition of all the SASP factors non-specifically by senolytics would also eliminate those reparative SASP factors involved in protective and reparative processes. For example, senescent murine cardiomyocytes release GDF15, TGFb2, and endothelin-3, contributing to cardiac fibrosis and hypertrophy [24]. On the other hand, CCN1 released from the cardiomyocyte induces fibroblast senescence, reduces cardiac fibrosis, and promotes cardiomyocyte regeneration and repair [63]. Therefore, senomorphics that specifically target destructive SASP factors while preserving reparative ones shall be a research focus. Because targeting individual factors will not be efficient, senomorphics, which target the common mechanisms for the production of a group of SASP factors, shall be of great interest. For example, inhibition of the mTOR pathway, which has been shown to extend the lifespan and reduce cell senescence phenotypes in aged mice [83], is able to reduce cardiac hypertrophy and improve cardiac function in aging mouse models [84]. Other interesting senomorphics, such as Sirt1 and AMPK activators [85], as well as senomorphics that regulate senescent cell metabolisms and, in turn, SASP factors, are of great interest. For example, pyruvate kinase isoenzyme 2 (PKM2) has been shown to accumulate and form aggregates in senescent cells, which diverts physiological glycolytic flux and drives cellular senescence. Compounds that are capable of dissolving PKM2 aggregates alleviate cell senescence and extend the lifespan of naturally and prematurely aged mice [86]. It is expected that novel mechanisms that regulate multiple SASP factors will be explored and that senomorphics that efficiently modulate these SASP factors will be developed in time in the near future.

Moreover, the SASP factor profiles from senescent cells might be different depending on the stimuli that induce cell senescence [87]. Replicative senescent vascular endothelial cells or the endothelium from aged mouse exhibit a pro-inflammatory phenotype [15], while the *SENEX* gene induced premature endothelial senescence, revealing an anti-inflammatory phenotype [73]. These observations implicate that senescent cells in different pathological conditions, such as atherosclerosis, insulin resistance, diabetes, hypertension, etc., and natural aging may have different profiles of SASP factors, which may also influence the effects of senolytic and/or senomorphic therapies.

Third, the concept that senescent cells are permanently cell-cycle-arrested is challenged by studies showing that senescent cells can reenter the cell cycle and resume normal function [88]. Reversing the proliferative capacity of senescent cells through exosomal miR-302b treatment in aged mice showed decreased senescent cell accumulation in multiple organs, increased body weight, and reversal of several aging phenotypes, including inhibition of chronic systemic inflammation, restoration of hair growth, muscle strengthening, improvement of cognition, and extended lifespan [88]. This so-called senoreverse strategy could be an interesting therapeutic modality for future research exploring anti-aging therapy. The reversal of cell cycle arrest has also been shown in the liver fibrosis model, in which overexpression of VEGFR2 in p16^ink4+^ endothelial cells increases their proliferation and blood vessel formation and reduces liver fibrosis [65].

Fourth, the drawback of currently developed senolytics, which eliminate senescent cells non-discriminately, shall be considered. The development of saver and more specific senolytic drugs is an important aspect of clinical use. The currently developed most popular senolytics are drugs that induce senescent cell death by targeting the anti-apoptotic pathways, such as BcL family proteins, which are upregulated in senescent cells or inhibitors of tyrosine kinase, which support senescent cell survival [89]. Side effects of the senolytics, such as toxicity to blood cells, including platelets and leukocytes, i.e., thrombocytopenia and leukocytopenia [90], and even cardiotoxicity effects are observed [8,90,91]. This is not too surprising, because those drugs are originally used as anti-cancer therapeutics, which exhibit significant cytotoxic effects. Indeed, ABT-263 treatment of atherosclerotic mice could also remove some non-senescent cells and may contribute to detrimental effects in this mouse model [71]. Preclinical studies with senolytics shall be performed in the old or at least adult mouse model with more advanced atherosclerosis to examine therapeutic instead of preventive effects, which more closely matches the clinical situation. In addition, more appropriate animal models of atherosclerotic plaque rupture shall be developed, because mice do not develop plaque rupture. Finally, senolytic therapy could be detrimental in the organs in which senescence cells are essential for the maintenance of tissue or organ homeostasis. For example, in the liver, in which senescent hepatic sinusoidal endothelial cells are important for vascular integrity, in the removal of senescent endothelial cells, they cannot be replaced by newly regenerated endothelial cells. As a consequence, the senolytic approach has been shown to cause perivascular fibrosis in the liver [92]. In the lung, fibroblasts with certain senescent characteristics in the basement membrane adjacent to epithelial stem cells exert a function of supporting epithelial regeneration, likely through their increased secretory capacity of SASP factors [93]. The tissue regeneration stimulating effects of senescent cells are also required for efficient healing; therefore, the depletion of senescent cells impairs wound closure in the skin [94]. Moreover, senescent pancreatic β-cells in aging play an important role in increasing insulin production in the elderly [95,96]. Removal of these senescence cells through senolytic therapy could be detrimental. The concept of the roles of senescent cells in cardiac diseases and atherosclerotic vascular disease and the effects of senolytics/senomorphics drugs on cardiovascular diseases are summarized and illustrated in Figure 1. These aspects shall be analyzed when studies with senolytics are performed. 

## 5. Conclusions

Anti-senescence therapies or senolytic and senomorphic drug therapies are attractive and promising future therapeutic modalities for the treatment of cardiovascular diseases, and they represent a novel research direction for cardiovascular aging and age-associated cardiovascular diseases. However, we shall be cautious not to endorse clinical use of senolytics for the prevention or treatment of cardiovascular diseases or other age-associated chronic diseases until the roles of cell senescence in specific disease development are well-studied and the safety and efficacy of the drugs in well-designed clinical trials are investigated.

## Figures and Tables

**Figure 1 biomolecules-15-00545-f001:**
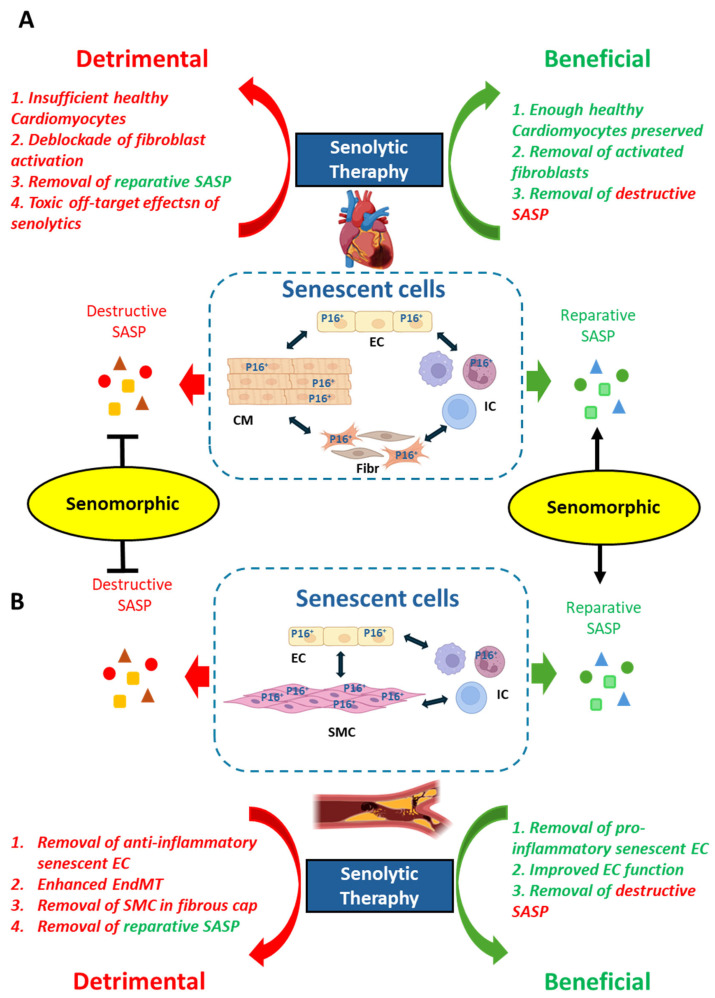
Illustration of the mechanisms of beneficial and detrimental effects exerted by anti-senescence therapy in heart disease (**A**) and atherosclerotic vascular disease (**B**). Anti-senescence therapies include senolytic and senomorphic therapy. While senolytic agents selectively eliminate senescent cells, senomorphic compounds eliminate the detrimental effects of senescent cells by targeting SASP (suppressing destructive while preserving reparative SASP). Details are discussed in the text of the article. EC, endothelial cell; CM, cardiomyocyte; IC, immune cell; Fibr, fibroblast; SMC, smooth muscle cell; SASP, senescence-associated secretory phenotype; EndMT, endothelial–mesenchymal transition.

## Data Availability

No new data were created.

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
