# Peer review of "Is Senolytic Therapy in Cardiovascular Diseases Ready for Translation to Clinics?"

_biomolecules, 2025, doi:10.3390/biom15040545_

Round 1

Reviewer 1 Report

Comments and Suggestions for Authors

This is a concise and timely contribution that discusses the controversies and conflicting findings regarding the use of senolytic approaches to clear senescent cells (of various types) in the context of cardiovascular disease. The authors very nicely present both sides of the issue and highlight the positive as well as negative aspects of anti-senescence targeting in the cardiovascular system. This important paper could benefit from a more detailed discussion of the benefit vs pathophysiology of the various SASP factors that are expressed in cardiomyocytes, macrophages and endothelial cells as they are very different and are dependent on the etiology of the various senescence states and whether involved cells are in early vs. deep senescence.. This does not have to be an all inclusive effort but some recognition that specific, perhaps even cell-type related, effectors of the senescent phenotype might underlie the different outcomes seen in senolytic therapy would be a useful inclusion. 

Author Response

This is a concise and timely contribution that discusses the controversies and conflicting findings regarding the use of senolytic approaches to clear senescent cells (of various types) in the context of cardiovascular disease. The authors very nicely present both sides of the issue and highlight the positive as well as negative aspects of anti-senescence targeting in the cardiovascular system. This important paper could benefit from a more detailed discussion of the benefit vs pathophysiology of the various SASP factors that are expressed in cardiomyocytes, macrophages and endothelial cells as they are very different and are dependent on the etiology of the various senescence states and whether involved cells are in early vs. deep senescence.. This does not have to be an all inclusive effort but some recognition that specific, perhaps even cell-type related, effectors of the senescent phenotype might underlie the different outcomes seen in senolytic therapy would be a useful inclusion. 

We would like to thank the reviewer's positive comments on our manuscript! Your suggestion to include more detailed discussion on various SASP factors expressed in cardiomyocytes, macrophages, and endothelial cells is well taken. In the literature, there are numerous review articles that have comprehensively summarized and illustrated SASP factors from different cell types in cardiovascular system in very much details. Considering that this is a "Opinion" article which discusses the most important or "burning" questions and controversies in the field of anti-senescence therapy in cardiovascular aging and disease, we would like to keep the discussion concise and focused. Therefore, we referred to those references and mentioned a few SASP factors as examples as suggested. In addition, strategies that targeting upstream mechanisms that regulate a group of SASP factors instead individual factors, which shall be more efficient for senomorphic therapy are discussed. Please see the revision on this aspect on page 6, line 257 - 284. 

As suggested, we also included a discussion on possible influence of early and deep persistent senescence on the senolytic therapy on page 5, the 1st paragraph, lines 224-226. Finally, the functions of senescent cells are dependent on the stimuli or stressors or risk factors are different, which may also affect the effects or the outcomes of of anti-senescence therapy. This aspect is also included on page 6, 2nd paragraph, lines 285 to 292. According to editor's comment, we have a separate paragraph for "Conclusion". 

I hope that the revision merits the reviewer's expectation. 

Sincerely yours

Zhihong Yang      

Reviewer 2 Report

Comments and Suggestions for Authors In this manuscript the authors discuss a very current topic, namely the possibility
of treating cardiovascular diseases through senolytic therapy. The article is very
interesting and well written. However, the authors neglected to address the role of
hyperinsulinemia associated with insulin resistance in determining cellular
senescence at various levels of the organism and, in particular, at the
cardiovascular level. It is known that the condition of
insulin resistance/hyperinsulinemia, whose prevalence is increasing enormously
in developed and developing countries, determines, if left untreated,
evident cardiovascular alterations with increased mortality.

Author Response

In this manuscript the authors discuss a very current topic, namely the possibility of treating cardiovascular diseases through senolytic therapy. The article is very interesting and well written. However, the authors neglected to address the role of hyperinsulinemia associated with insulin resistance in determining cellular senescence at various levels of the organism and, in particular, at the cardiovascular level. It is known that the condition of insulin resistance/hyperinsulinemia, whose prevalence is increasing enormously in developed and developing countries, determines, if left untreated, evident cardiovascular alterations with increased mortality.

We would like to thank the reviewer's positive comments on our manuscript! Your suggestion to include the aspect of insulin resistance/hyperinsulinemia and cardiovascular aging is well taken. Indeed, the functions of senescent cells are different and dependent on the stimuli or stressors and perhaps also on the risk factors. For example, replicative senescent vascular endothelial cells or the endothelium from aged mouse exhibit a pro-inflammatory phenotype [15], while SENEX gene induced premature endothelial senescence reveal an anti-inflammatory phenotype [73]. These observations implicate that senescent cells in different pathological conditions such as atherosclerosis, insulin resistance, diabetes, hypertension, etc, and natural aging may have different profiles of SASP factors, which may also influence the effects of senolytic and/or senomorphic therapies. We think that one has to analyse the mechanisms and SASP profiles in details in diabetes and insulin resistance and compare with the those in other cardiovascular risk factors. To our knowledge, no such studies or systematic analyses have been done. We feel that this aspect warrants thorough investigation and can not be well addressed in the present article of the "Opinion" format which discusses the most important or "burning" questions and controversies in the field of anti-senescence therapy in cardiovascular aging and disease. A specific review article devoted only to the effects of insulin resistance on cardiovascular aging and SASP factors and anti-senescence therapy under this condition is required. Nevertheless, we have included a paragraph on page 6, 2nd paragraph, lines 285 to 292 to point out this aspect and feel that it is better to incorporate the insulin resistance aspect with other risk factors.  

According to editor's comment, we have a separate paragraph for "Conclusion".  Answers to reviewer Nr. 1 is highlighted in red in the revision. 

I hope that the revision merits the reviewer's expectation. 

Sincerely yours

Zhihong Yang      

Round 2

Reviewer 2 Report

Comments and Suggestions for Authors

The authors have correctly answered to my question, and I believe that the manuscript can be accepted for publication in the Journal in ita current form.